# Schwannoma: A Rare Case of Submucosal Gastric Tumor

**DOI:** 10.3390/diagnostics13122073

**Published:** 2023-06-15

**Authors:** Cosmina Fugărețu, Cătalin Mișarca, Lucian Petcu, Raluca Șoană, Andrada Cîrnațiu, Marin Valeriu Surlin, Stefan Patrascu, Sandu Ramboiu

**Affiliations:** 11st General Surgery Department, Brașov County Emergency Clinical Hospital, 500326 Brașov, Romania; 2Faculty of General Medicine Brașov, Transilvania University, 500036 Brașov, Romania; 3Pathological Anatomy Department, Brașov County Emergency Clinical Hospital, 500326 Brașov, Romania; 41st General Surgery Department, Emergency Hospital of Craiova, 200642 Craiova, Romania; 5Faculty of General Medicine Craiova, University of Medicine and Pharmacy of Craiova, 200349 Craiova, Romania

**Keywords:** gastric schwannoma, submucosal gastric tumors, rare gastric tumors

## Abstract

Schwannoma is a tumor that originates from the Schwann cells that surround a neuron’s axon. This tumor is very rare in the gastrointestinal tract and develops submucosally from intestinal nerve plexuses. The most common location for gastrointestinal schwannomas is the stomach, where they account for only 0.2% of gastric tumors. We present the case of a 56-year-old asymptomatic patient who was diagnosed, following a routine ultrasound examination, with an abdominal tumor. An abdominal MRI confirmed the gastric origin of the tumor. Although a subsequent upper-digestive endoscopic ultrasound was performed, a definitive diagnosis could not be established. Thus, a laparoscopic wedge resection of the stomach was performed. The immunohistochemical examination of the tumor established the diagnosis of benign schwannoma. Despite the availability of advanced endoscopy and imaging techniques, the diagnosis of gastric schwannoma is very rarely preoperative. The immunohistochemical identification of S-100 on the surgical specimen confirmed the diagnosis.

**Figure 1 diagnostics-13-02073-f001:**
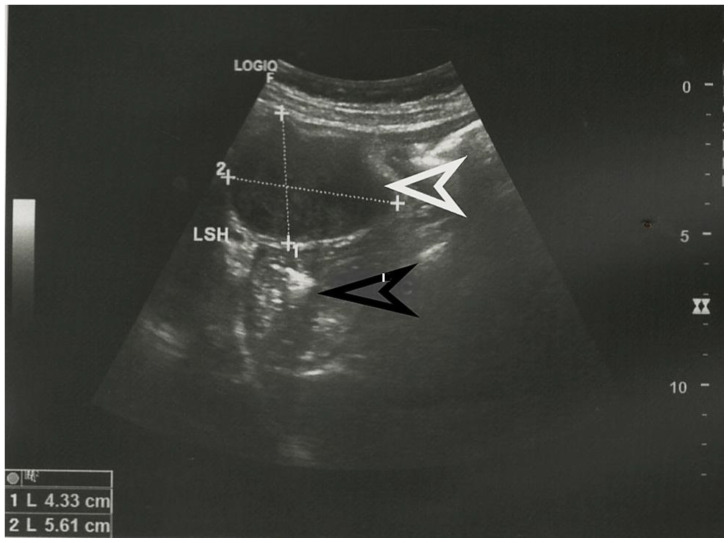
Abdominal ultrasound showing the presence of an iso-hypoechoic tumoral formation, marked with a white arrow, in a 56-year-old patient without any abdominal symptoms. The stomach is indicated with a black arrow and the left hepatic lobe is marked LSH. The dimensions of the tumor in cm are specified at the bottom left. No other pathological changes were identified by ultrasound. Submucosal tumors, also known as mesenchymal tumors, are a group of tumors that contain spindle-shaped cells [1]. They are classified histopathologically as a gastrointestinal stromal tumor (GIST), tumors originating from smooth muscle tissue, which, in turn, includes leiomyomas, leiomyosarcomas, and tumors originating from nervous tissue. The latter category includes schwannomas, granular cell tumors, and neurofibromas [2]. Gastric schwannoma is a tumor that originates from the nerve plexus of the intestinal wall [3]. More specifically, this tumor originates from Schwann cell sheaths and represents 0.2% of all gastric tumors [4]. Most commonly, gastric schwannoma originates from the Auerbach plexus, located in the smooth muscle layer of the gastric wall, and less commonly from the Meissner submucosal plexus. As it grows, the tumor pushes the nerve to the periphery, with no change in neuronal function [5]. This tumor is most often benign and is more common in women between the ages of 50–60 [4,6]. We present the case of a 56-year-old asymptomatic patient who was diagnosed, following a routine ultrasound examination, with an abdominal tumor. The ultrasound appearance of the tumor is presented in Figure 1.

**Figure 2 diagnostics-13-02073-f002:**
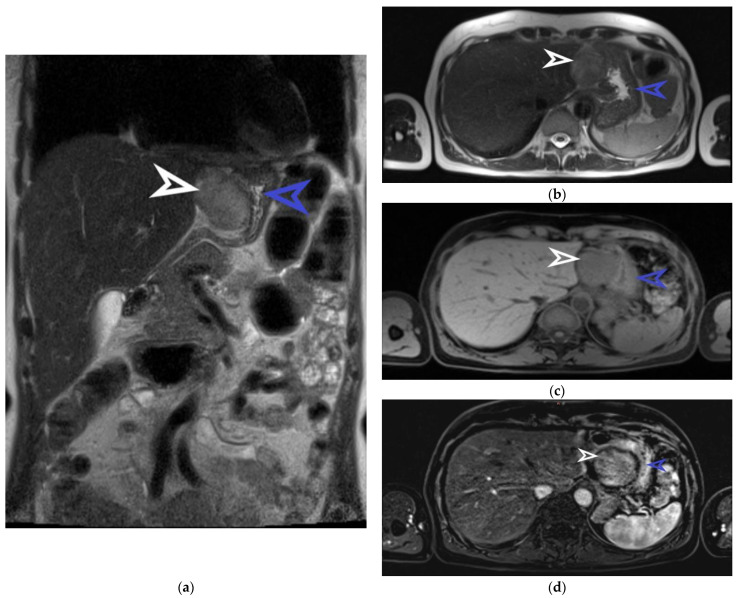
Abdominal MRI examination, coronal section. The well-defined tumor originating from the anterior gastric wall with a heterogeneous T2 intermediate signal is marked with a white arrow. The stomach is indicated by the blue arrow (**a**). Abdominal MRI examination, transverse section, and T2 sequence. The tumor is marked with a white arrow and the stomach is marked with a blue arrow (**b**). Abdominal MRI examination, transverse section, and T1 sequence. The tumor is indicated by the white arrow and the stomach is indicated by the blue arrow (**c**). Abdominal MRI examination, transverse section, and T1 arterial sequence. The tumor is marked with a white arrow and is shown to have a hypo-signal T1, gadofil, and slightly restricted diffusion. The blue arrow indicates the stomach (**d**). The patient was examined and treated in the Brașov County Emergency Clinical Hospital and in the Emergency Hospital of Craiova between November and December 2022. She is not on any medication at present nor on any proton pump inhibitors. She has been in menopause for a couple of years, had 2 natural births, and had her first menstruation at the age of 15. The laboratory exams did not reveal any pathological changes, as she had a hemoglobin value of 14.8 g/dL, red blood cells 5 × 10^6^/uL, white blood cells 6.54 × 10^3^, INR 1.01, APTT 21.7 sec, urea 43 mg/dL, and creatinine 0.86 mg/dL. The patient later underwent an abdominal MRI for the characterization and determination of the tumor’s origin. This imaging examination described the tumor detected by ultrasound as round-oval. It was found to have an exophytic growth from the gastric wall, dimensions of 44/50 mm, an intermediate heterogeneous T2/T2 FS signal, hyposemnal T1, to be gadophilic, and to overall be slightly restricted in diffusion. Figure 2 captures some suggestive images of the tumor’s appearance on the MRI.

**Figure 3 diagnostics-13-02073-f003:**
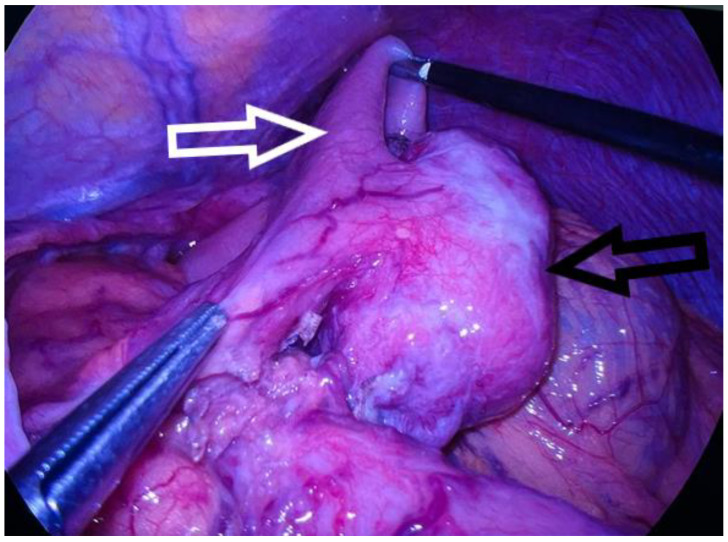
Intraoperative image. The well-delimited tumor that developed from the anterior stomach wall is marked with a black arrow, and the stomach is spread with the help of the two grasping forceps, marked with a white arrow. Based on the description of the MRI, there was a suspicion that the identified tumor was a gastrointestinal stromal tumor (GIST) and thus an upper-digestive endoscopy was recommended. The endoscopy was carried out shortly after and showed that the tumor was located in the submucosal layer of the lesser curvature of the stomach. The echo-endoscopy examination revealed the tumor’s hypoechoic appearance and origin from the muscle layer without causing disruption to the parietal stratigraphy. Two small tissue fragments were taken for histopathologic analysis and were described as having elongated nuclei, reduced nuclear pleomorphism, and an eosinophilic cytoplasm that may correspond to a mesenchymal type of tumor proliferation. However, an immunohistochemical determination of CD 117, CD34, desmin, actin, and S-100 was necessary to confirm the results. Due to the small size of the tissue collected through eco-endoscopic guidance, immunohistochemical examination could not be performed and the patient was admitted to the surgery clinic for a wedge gastrectomy. Endoscopic resection was excluded because the tumor size exceeded 3 cm. Thus, laparoscopic surgery was performed and a 5/5 cm encapsulated gastric tumor was discovered, originating from the anterior gastric wall near the lesser curvature; no other pathological findings of the intra-abdominal organs were detected. A partial resection of the anterior gastric wall was performed using an Endo-GIA purple stapler. The intraoperative perspective can be seen in Figure 3.

**Figure 4 diagnostics-13-02073-f004:**
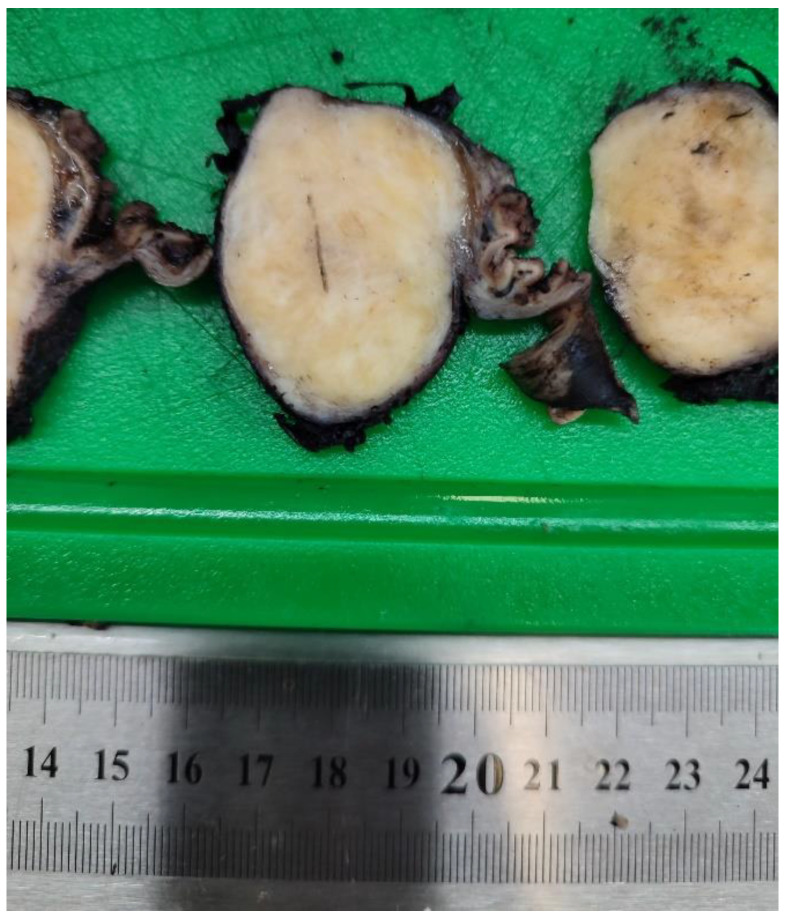
Macroscopic image of tumor formation after fixation and sectioning. The round-oval shape of the well-defined tumor is visible. Post-surgery, the patient received symptomatic treatment and a proton pump inhibitor. She was allowed to drink liquids and semi-solids from the first and second postoperative days. The nasogastric tube was removed on the second postoperative day. The patient’s evolution was favorable, with the passage of flatus and stools being present from the second postoperative day. She was discharged in good general condition on the fourth postoperative day. The macroscopic histopathological examination of the wedge-type gastric resection specimen describes a well-defined tumoral formation of 50/42/31 mm. The macroscopic appearance of the tumor after the piece had been fixed and sectioning is presented in Figure 4.

**Figure 5 diagnostics-13-02073-f005:**
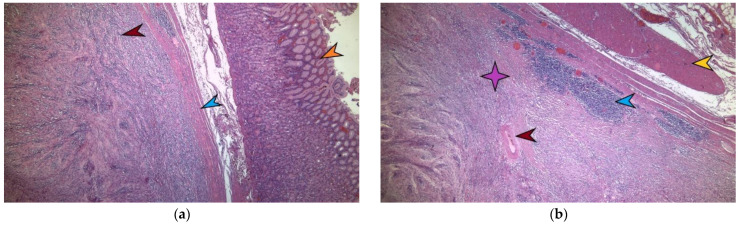
A microscopic image of tumor formation in hematoxylin–eosin stain, ×4 magnification. The red arrow marks the tumor formation originating from the stomach’s own muscular layer. The blue arrow indicates the stomach’s own muscular layer, and the yellow arrow indicates the normal-looking gastric mucosa above it (**a**). A detailed microscopic image of a gastric schwannoma in hematoxylin–eosin stain, ×10. The purple star indicates the tumoral mass made of monomorphic fusiform cells with eosinophilic cytoplasm and elongated nuclei, where no mitotic activity and no intratumoral necrosis are identified. The red arrow indicates an intratumoral blood vessel with hyalinized walls. At the periphery of the tumor, lymphoid-cuffing aggregates are evident, marked by the blue arrow. The yellow arrow marks the muscularis propria of the stomach (**b**). Microscopically, in hematoxylin–eosin staining, a well-delimited nodular tumor proliferation is detected and develops at the level of the gastric muscle proper. This tumor consist of monomorphic fusiform cells with an eosinophilic cytoplasm and elongated nuclei, with no mitotic activity identified, and with no intratumoral necrosis (Figure 5a,b). Thus, the exact type of tumor is determined by the immunohistochemical determination of CD117, CD 34, SMA—smooth muscle actin, and desmin—which are negative in this case. CD 117 is a marker that is present in gastrointestinal stromal tumors (GIST) and is expressed by the c-kit tyrosine kinase receptor and Cajal cells in the intestine [7].

**Figure 6 diagnostics-13-02073-f006:**
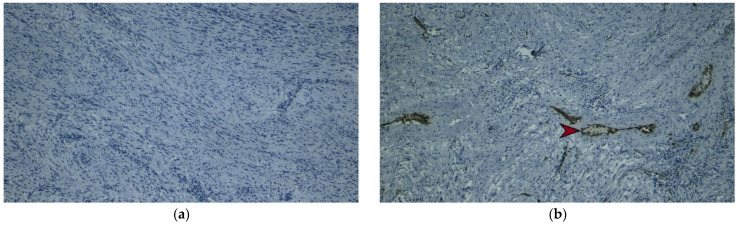
Immunohistochemical image, ×10, of the gastric tumor formation showing the absence of CD 117 expression in the tissue proliferation (**a**). Immunohistochemical image, ×10, which confirms the absence of CD 34 expression in the tumor, which is present at the level of the vascular endothelium marked with a red arrow (**b**).

**Figure 7 diagnostics-13-02073-f007:**
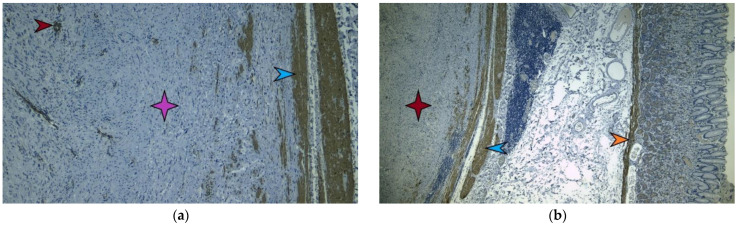
Immunohistochemical image, ×10, which investigates the presence of SMA (smooth muscle actin). The tumor tissue is marked with a purple star and is negative for SMA. The presence of SMA is evident in the smooth muscle layer of intratumoral vessels indicated by the red arrow and in the proper muscle layer of the stomach indicated by the blue arrow (**a**). Immunohistochemical image, ×4, showing the absence of desmin in the tumor tissue. The absence is marked with a red star and its presence in the smooth muscle layer of the stomach is indicated by a blue arrow, just as in the muscle layer of the mucosa, where it is indicated by a yellow arrow (**b**).

**Figure 8 diagnostics-13-02073-f008:**
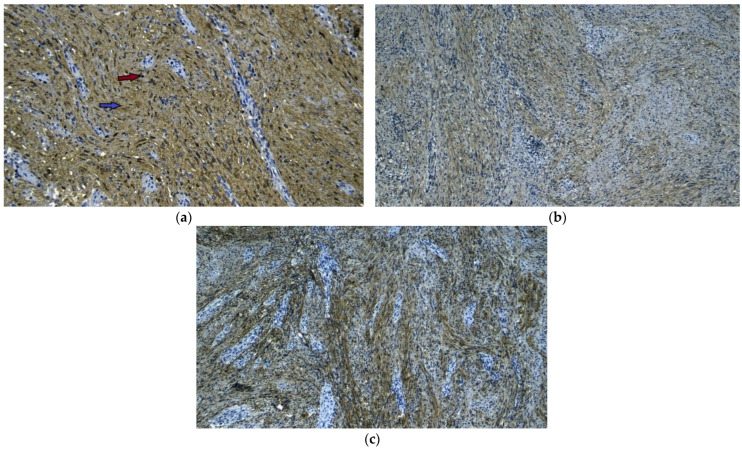
Immunohistochemical image, ×10, showing the presence of S-100 in large amounts in the tumor, both at the cellular nucleus indicated by the red arrow and in the cytoplasm indicated by the blue arrow. S-100 is a protein that is normally only found in Schwann cells originating from the neural crest, so we can establish the diagnosis of schwannoma (**a**). Immunohistochemical image, ×10, showing positive staining for vimentin in the tumor (**b**). Immunohistochemical image, ×20, showing the presence of glial fibrillary acidic protein (GFAP) in the tumor, indicating its origin from the myenteric plexus in the schwannoma (**c**). Thus, the absence of positivity for CD117 and CD 34 eliminates the diagnosis of GIST (Figure 6a,b). The absence of tumor expression of SMA—smooth muscle actin and desmin—obliges us to eliminate the diagnosis of leiomyoma (Figure 7a,b). The strong expression of S-100, a protein that is normally found only in Schwann cells originating from the neural crest, allows us to establish the diagnosis of schwannoma (Figure 8a). Vimentin is also diffusely positive in the tumor (Figure 8b). The glial fibrillary acidic protein (GFAP), which indicates an origin from the myenteric plexus, is positively zonal in tumor growth (Figure 8c). Thus, the diagnosis of gastric schwannoma is established without a doubt. Gastric schwannoma has a slow growth rate, and patients are often asymptomatic or have non-characteristic manifestations such as epigastric pain, episodes of upper-digestive bleeding, and even weight loss in 10–25% of cases [2]. The most common location for gastrointestinal schwannomas is the stomach [8]. Most gastric schwannomas are located in the body of the stomach, followed by the antrum, and the gastric fundus is the least common [4]. However, tumor positioning at small gastric curvatures seems to be the most common [9]. Among the useful paraclinical investigation methods in the diagnosis of gastric schwannoma, upper-digestive endoscopy is the first choice. The most frequently seen endoscopic image is a submucosal protuberance with normal mucosal cover. However, in one-quarter to one-half of the cases, we find central mucosal ulceration [10]. Although most cases of schwannoma are benign tumors, there are several cases of malignant schwannomas that have been reported [9]. This malignant character cannot be established clinically or through imaging examinations, although some authors argue that FDG (18F-fluorodeoxyglucose) PET-CT examination can distinguish between malignant cases that exhibit increased FDG uptake [9,11]. The presence of adjacent lymphadenopathies could also raise the suspicion of malignancy [12]. However, increased FDG uptake has also been observed in benign schwannomas, which could be explained by the role of Schwann cells in glucose transport being necessary for axonal repolarization [13]. No correlation has been found between tumor size, ki-67 index, and FDG uptake rate, making histopathological examination essential for both establishing the diagnosis method of schwannoma and for characterizing it as benign or malignant [13]. Although there are no certain indicators of the malignancy of digestive schwannomas, a MIB-1 ki-67 proliferative index higher than 10% and a mitotic activity rate above 5 HPF, along with nuclear atypia and tumor dimensions over 5 cm, are associated with an increased risk of malignancy [6]. There are even cases described in the literature where patients developed secondary liver determinations, leading to death [9]. In the case presented, it is clear that we are faced with benign schwannoma, as no mitotic activity was identified, and the ki-67 is 2%. Most often, schwannomas are confused with GISTs due to their submucosal location and well-defined, encapsulated appearance [3]. Immunohistochemical examination shows a strong expression of S-100 in schwannomas compared to GISTs, which are positive for CD-34, c-kit, and DOG 1 or leiomyoma, which is positive for desmin and smooth muscle actin (SMA) [2,14]. Immunohistochemical positivity for glial fibrillary acidic protein (GFAP) indicates an origin from the enteric plexus; thus, these tumors can be considered to be different to peripheral schwannomas [8]. The removal of submucosal gastrointestinal tumors is recommended when the tumor size is over 2 cm. Tumors below this size have an indication for endoscopic monitoring [15]. The endoscopic removal of the tumor is recommended for tumors under 3 cm and can be performed through endoscopic submucosal excavation (ESE), endoscopic full-thickness resection (EFTR), and submucosal tunneling endoscopic resection [15,16]. Endoscopic removal of gastric schwannoma has also been found to be useful and safe, with good long-term results. Tumors as large as 55 mm have been removed endoscopically [16]. The surgical removal of schwannoma, either through a classical or laparoscopic approach, with gastric resection or limited but complete excision of the tumor, is associated with good long-term results without significant recurrence rates [17]. However, some authors have observed the concurrent presence of malignant diseases such as gastric cancer, colon cancer, renal cancer, non-Hodgkin’s lymphoma, GIST, breast cancer, pancreatic cancer, and lung cancer. In 50% of cases, the cancer diagnosis was made before the discovery of the schwannoma, in 40% it was made concurrently, and in only one case it was made after the discovery of the benign tumor [6]. In conclusion, the diagnosis of gastric schwannoma is very rarely established preoperatively, despite the advanced diagnostic arsenal offered by echo-endoscopy, CT, or PET [5]. The removal of the tumor through endoscopic or surgical procedures, adapted to each case, allows for the accurate establishment of the diagnosis.

## Data Availability

The data presented in this study are available on request from the corresponding author.

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
