# Peer review of "Schwannoma: A Rare Case of Submucosal Gastric Tumor"

_diagnostics, 2023, doi:10.3390/diagnostics13122073_

Round 1
Reviewer 1 Report
Dear Authors
Thank you for your manuscript submission. Your work is interesting; however, a Minor Revision is needed as below:
1. Please do mention the place of the study (City, Country) and duration of the study (time of the beginning MM/DD/YYYY- time of the end MM/DD/YYYY)
2. You can compact your figures; for example compact four figures within a one and show them as a/b/c/d
3. It is recommended to add a flow chart to show all the applied procedures in your manuscript.
4. Please do write each term in its complete form for the first time and then use its abbreviation e.g., GIST (p. 3), etc.
5. It is recommended to read and add the following papers to References section of the manuscript to have fruitful manuscript:
Clinical Diagnosis of Gastrointestinal Stromal Tumor (GIST): From the Molecular Genetic Point of View. Cancers 2019, 11, 679. https://doi.org/10.3390/cancers11050679
Preoperative Diagnosis Failure for a Rare Gastric Collision Tumor: A Case Report. Diagnostics 2021, 11, 633. https://doi.org/10.3390/diagnostics11040633
Recent Progress and Challenges in the Diagnosis and Treatment of Gastrointestinal Stromal Tumors. Cancers 2021, 13, 3158. https://doi.org/10.3390/cancers13133158
Gastrointestinal Stromal Tumors Mimicking Gynecologic Disease: Clinicopathological Analysis of 20 Cases. Diagnostics 2022, 12, 1563. https://doi.org/10.3390/diagnostics12071563
Gastrointestinal Stromal Tumors (GIST): A Population-Based Study Using the SEER Database, including Management and Recent Advances in Targeted Therapy. Cancers 2022, 14, 3689. https://doi.org/10.3390/cancers14153689
Reviewer 2 Report
This is a rare and interesting case. The content written by the author is too verbose and needs to be more concise.
Reviewer 3 Report
Hi Editor and Author,
I am honored to review the candidate report titled as "Schwannoma: a rare case of submucosal gastric tumor".
While rare case report of a clinical disease is of significance; the evidence and readout is solid and conclusion is clear, I would recommend a major reversion of this case report by following reasons:
1. The language is not proper. There are lots of grammar error, including time tense, together with broken sentence that are difficult to understand. The language of this paper must be improved professionally.
2. The paper is not organized in the academic manner. Please read the following instructions for manuscript preparation.
a. https://www.ncbi.nlm.nih.gov/pmc/articles/PMC5686928/
b. https://diagnosticpathology.biomedcentral.com/submission-guidelines/preparing-your-manuscript/case-report
In summary, clinical case report is of interest to general readers, however extensive improvements in manusript need to be done before access to general readers.
Thanks!
Round 2
Reviewer 3 Report
N/A
